# Biocompatibility of ZrO_2_ vs. Y-TZP Alloys: Influence of Their Composition and Surface Topography

**DOI:** 10.3390/ma15134655

**Published:** 2022-07-01

**Authors:** Alex Tchinda, Laëtitia Chézeau, Gaël Pierson, Richard Kouitat-Njiwa, B H Rihn, Pierre Bravetti

**Affiliations:** Jean Lamour Institute, University of Lorraine, UMR 7198, 54011 Nancy, France; laetitia.chezeau@univ-lorraine.fr (L.C.); gael.pierson@univ-lorraine.fr (G.P.); richard.kouitat@univ-lorraine.fr (R.K.-N.); bertrand.rihn@univ-lorraine.fr (B.H.R.); pierre.bravetti@gmail.com (P.B.)

**Keywords:** osseointegration, biocompatibility, zirconia, yttria–zirconia, surface topography, proliferation, morphology, transcriptome

## Abstract

The osseointegration of implants is defined as the direct anatomical and functional connection between neoformed living bone and the surface of a supporting implant. The biological compatibility of implants depends on various parameters, such as the nature of the material, chemical composition, surface topography, chemistry and loading, surface treatment, and physical and mechanical properties. In this context, the objective of this study is to evaluate the biocompatibility of rough (Ra = 1 µm) and smooth (Ra = 0 µm) surface conditions of yttria–zirconia (Y-TZP) discs compared to pure zirconia (ZrO_2_) discs by combining a classical toxicological test, morphological observations by SEM, and a transcriptomic analysis on an in vitro model of human Saos-2 bone cells. Similar cell proliferation rates were observed between ZrO_2_ and Y-TZP discs and control cells, regardless of the surface topography, at up to 96 h of exposure. Dense cell matting was similarly observed on the surfaces of both materials. Interestingly, only 110 transcripts were differentially expressed across the human transcriptome, consistent with the excellent biocompatibility of Y-TZP reported in the literature. These deregulated transcripts are mainly involved in two pathways, the first being related to “mineral uptake” and the second being the “immune response”. These observations suggest that Y-TZP is an interesting candidate for application in implantology.

## 1. Introduction

For many years, commercially pure titanium and its alloys have been the gold standard in oral implantology. Over the last century, many improvements have occurred and different materials have emerged, such as resins and ceramics. The most common types of dental implant materials are titanium and zirconia. Titanium and its alloys remain widely used due to their many advantages, such as their excellent biocompatibility, high corrosion resistance, passivation capacity, and excellent mechanical properties [1,2]. However, one limitation is the prevalence of peri-implantitis, an immune-mediated biological complication attributed to bacterial biofilms on the implant surface [3]. Gram-negative oral bacteria such as *Tannerella fosythia, Campylobacter gracilis,* and *Porphyromonas gingivalis* and Gram-positive bacteria such as *Streptococcus intermedius* and its *mutants* are involved in this disease. These species possess virulence factors that can lead to soft and hard tissue destruction [4]. The mucosal lesions induced by peri-implantitis are accompanied by a deep pocket with bleeding, suppuration, and marginal bone loss, which can result in osseointegration failure. The prevalence of peri-implantitis in a group of Moroccan patients (642 implants in 145 subjects followed up for a mean of 6.4 years) was 41.4% at the subject level and 22.7% at the implant level [5]. A study conducted by Tchinda et al. in 2021 suggested that anaerobic bacteria of the genus *Desulfovibrio fairfieldensis* can create biofilm colonies on titanium coupons and proliferate in vitro under oral physiological conditions [6]. Moreover, a few cases of titanium allergies involving dental implants are reported in the literature [7].

In recent years, zirconia dental implants have emerged as an alternative to titanium. Indeed, zirconia or zirconium dioxide (ZrO_2_) and its alloys have exceptional mechanical properties, esthetic outcomes, biocompatibility, and resistance to corrosion [8,9,10]. Moreover, discoloration or hypersensitive reactions with allergies are not observed contrary to titanium implants [11]. Zirconia is a glass-free polycrystalline ceramic. Its excellent mechanical properties are related to the concentration of the crystalline phase. The transition from the quadratic to the monoclinic phase during the shaping and cooling process from 1000 to 1100 °C leads to a volume increase of about 3%, which could be damaging to the structure of the material [12]. Thus, the addition of a small number of oxides such as magnesium oxide (MgO), calcium oxide (CaO), yttrium (Y_2_O_3_), and alumina (Al_2_O_3_) is necessary to stabilize zirconia in the tetragonal or cubic phase at room temperature. The main alloy obtained is yttria-stabilized tetragonal zirconia polycrystal (Y-TZP) [13]. Several variants of Y-TZP are available, depending on the additives and dopants, sintering profiles, and ensuing heat treatments [9]. In this study, Y-TZP refers to zirconia stabilized by the addition of 4 mol% yttrium oxide (Y_2_O_3_). Many in vivo (humans, rats, monkeys, rabbits) and in vitro (gingival fibroblasts, osteoblast-like cells, oral keratinocyte) studies report the excellent biocompatibility of ZrO_2_ and its alloys [14,15,16,17,18,19].

On the other hand, zirconia implants seem to induce less biofilm formation compared to titanium, resulting in less risk of peri-implantitis [20,21]. However, some recent studies suggest a similar effect of zirconium compared to titanium implants regarding its biocompatibility, osteoconductivity, physical properties, allergenicity, and biofilm formation [22,23,24,25,26].

As osseointegration is not only conditioned by the intrinsic nature of the materials, the surface chemistry and properties are essential in this process, influencing the protein adsorption and osteoblastic cell adhesion [27]. For instance, well-documented studies based on titanium implants have highlighted the importance of roughness to promote osseointegration, favoring both bone anchoring and biomechanical stability [28,29]. Similarly, surface modifications of zirconia-based implants could influence the osseointegration process and the adherence of bacteria [11,30,31]. Thus, surface modifications and treatments seem essential to improve the bone formation, biocompatibility, and limited biofilm formation. However, the precise role of the surface chemistry and topography in the early events of dental implant osseointegration remains poorly understood [28,32].

In this context, the present study assesses the biocompatibility of pure zirconia discs ZrO_2_ compared to yttria–zirconia discs (Y-TZP) as well as the influence of the surface topography through the combination of a conventional toxicological assay, morphological observations, and a transcriptomic analysis on an in vitro model of human Saos-2 bone cells. Although the biomechanical properties of zirconia and its alloys are widely documented in the orthopedic literature, only a few investigations have been carried out on the influence of ZrO_2_ on the gene expression profile, with none having been carried out on Y-TZP to our knowledge.

## 2. Material and Method

### 2.1. Preparation of ZrO_2_ and Y-TZP Discs

#### 2.1.1. Polishing Surfaces

ZrO_2_ and Y-TZP discs measuring 20 mm in diameter and 3 mm in thickness were obtained from Ampere Alloys^®^ (Saint-Ouen-l’Aumône, France).

The so-called “rough” surfaces had an arithmetic mean roughness Ra of about 1 µm. To obtain this surface condition, each face of the zirconia and yttria–zirconia discs was polished with a mechanical polisher (Struers^®^) for 1 min. P80 discs were also used (ESCIL^®^, Chassieu, France). The lubricant used was water supplied by the city of Nancy.

The so-called “mirror” surfaces had an arithmetic mean roughness Ra of about 0.04 µm. To achieve this surface finish, polishing with water lubrication was performed for each abrasive disc (P80, P180, P320, P500, and P600). The mirror-polish finish was achieved by polishing with 1 µm magnetic velvet cloths (Presi, Lot No:13151-18-1450, Saint-Étienne, France), which were moistened with absolute ethanol and continuously lubricated with a 2:3 suspension of hydrogen peroxide (Acros Organics, Lot: A0242019, Geel, Belgium) and OPS (Struers, Cat No.40700000, Ballerup, Denmark) for 3 min. This process was performed on the front and backside of each disc.

The chemical residues from the polishing process were cleaned in successive ethanol and osmosis water baths via ultrasonic sonication. The discs were then dried and autoclaved following the “prion” sterilization program at 134 °C and 2.1 Bar for two 18 min cycles for complete sterilization. The disks prepared in this manner were directly used for the cell culture.

#### 2.1.2. Surface Characterization with Tactile Profilometer

The measurement of the roughness of the discs after polishing and cleaning was carried out using a contact or tactile profilometer (Bruker, DektakXT stylus^®^, Institut Jean Lamour, Nancy, France). The accuracy of this device is of the order of plus or minus 1 nm.

The operation was performed according to ISO 4287:1997 standards with the following settings:i.Measurement length: 1000 μm;ii.Long cut-off: 0.8 μm; short cut-off: 0.08 µm.

The measurements were performed 3 times on each sample in different areas and then averaged.

#### 2.1.3. Scanning Electron Microscopy of Discs

After polishing and cleaning the disks, some were metalized by depositing a 15-nm-thick carbon layer in a metallizer (Safematic Compact Coating unit-010, Institut Jean Lamour, Nancy, France) under pressure for 10 min. The discs were then observed in a Quanta™ FEG 650 SEM (Institut Jean Lamour, Nancy, France) at 5.00 kV.

### 2.2. Cell Culture

The Saos-2 bone cells derived from an osteosarcoma in an 11-year-old Caucasian girl. This cell line is an ethically established and recognized line, which we obtained directly from the American Type Culture Collection (ATCC^®^ HTB85™). These cells were cultured with McCoy’s modified 5A medium (Gibco™ Paisley, UK) supplemented with 1% penicillin–streptomycin, 0.05% amphotericin B, and 15% fetal bovine serum (FBS, Sigma-Aldrich^®^, Saint Quentin Fallavier, France) and incubated in a humid atmosphere (37 °C, 5% CO_2_). The medium was refreshed every 2–3 days.

### 2.3. Cell Proliferation

The cell proliferation was measured using the WST-1 assay (Roche Applied Science, Mannheim, Germany), based on the reduction of the tetrazolium salt WST-1 to formazan by mitochondrial dehydrogenases in metabolically active cells. Saos-2 cells grown at passage 21 were seeded in 12-well plates at a density of 3.5 × 10^4^ cells/mL (2 mL per well) onto ZrO_2_ and Y-TZP discs with rough (Ra = 1µm) and mirror-polished (Ra = 0.04 µm) surface conditions, or onto the original plastic dish (control). The cells were cultured in a humid atmosphere (37 °C, 5% CO_2_) for 24, 72, and 96 h in the presence of discs (N = 4 independent experiments). At each time, the medium was removed, the cells were washed twice, and 200 µL (10% of the final well volume) of WST-1 reagent was added to each well containing 2 mL of cell suspension. The cells were incubated for 4 h at 37 °C and 5% CO_2_. A technical triplicate of 170 µL of cell suspension was made from each well and placed in a new 96-well plate The absorbance was measured at 450 nm using a microplate reader (FLUOstar^®^ Omega, BMG Labtech, Institut Jean Lamour, Nancy, France).

Different controls were performed, namely medium alone and medium with each disc, to ensure the absence of interference between the culture medium, the discs, and the WST-1 reagent.

Statistical analyses were performed by comparison of means using a one-way ANOVA. The results were considered significant when the “*p*-value” was less than 0.05 (*).

### 2.4. Scanning Electron Microscopy of Saos-2 Cells Exposed to Discs

The morphologies of the Saos-2 cells seeded at 2.5 × 10^4^ cells/well at passage 27 onto ZrO_2_ and Y-TZP discs with rough (Ra = 1 µm) and mirror-polished (Ra = 0.04µm) surface conditions during 96 h were observed by SEM. The cells were fixed with 5% glutaraldehyde (Sigma-Aldrich^®^, Lot:#SLCC3121, St. Louis, MO, USA) at 4 °C for 30 min, rinsed 3 times with phosphate buffer (Sigma-Aldrich^®^, Lot:#SLBZ5107, St. Louis, MO, USA), and then resuspended in 1% osmium (Sigma-Aldrich^®^, CAS:20816-12-0, St. Louis, MO, USA) for 30 min. The samples were then dehydrated by a concentration gradient of ethanol in distilled water (35%, 50%, 75%, 95%, 95%, 100%, 100% (*v*/*v*)) (Sigma-Aldrich^®^, St. Louis, MO, USA) for 15 min each time. Then, the samples were chemically dehydrated using hexamethyldisilane (Sigma-Aldrich^®^, Lot:#SHBJ6186, St. Louis, MO, USA) overnight in a vacuum desiccator (12 h). After complete drying, the samples were metalized for 10 min by gold sputtering to obtain a gold layer of ~15 nm (Safematic Compact Coating unit-010, Institut Jean Lamour, Nancy, France) and observed with the FEI Quanta 650 FEG™ SEM (Institut Jean Lamour Nancy, France).

### 2.5. Total RNA Extraction and Purification

The Saos-2 bone cells at passage 24 were seeded in 12-well plates at 3 × 10^4^ cells/mL (2 mL/well) onto rough Y-TZP discs or onto the original plastic dish (control) for 96 h. For each disc, 4 independent sets of experiments were performed. For each experiment, a total of 1.8 × 10^5^ cells seeded in 3 wells (6 × 10^4^ cells per well) were used to provide a sufficient amount of total RNA for microarray hybridization. The cells were trypsinized, centrifuged, and then lysed with RLTplus™ buffer containing β-mercaptoethanol (Qiagen^®^ Lot N°157031462, Hilden, Germany). The total RNA was extracted and purified using an RNeasy Plus Mini Kit from Qiagen^®^ (#74136, Hilden, Germany). The RNA purity was assessed via spectrophotometry using a BioSpecNano^®^ instrument (Shimadzu, Institut Jean Lamour, Nancy, France). The RNA quality and integrity were assessed via microfluidic electrophoresis using an RNA 6000 Nano Kit and 2100 Bioanalyzer (Agilent, Institut Jean Lamour, Nancy, France). All RNA samples were of high purity and integrity, as demonstrated by their A_260_/A_280_ ratios being above 2, with the RNA integrity numbers (RINs) varying from 9 to 9.5. The total RNA was stored at −80 °C until further use.

### 2.6. Microarray Hybridization

An aliquot (100 ng) of RNA from each sample was labeled with Cyanine 3-CTP using Low-Input Quick Amp Labeling kits (Agilent Technologies, Waldbronn, Germany). Labeled cRNAs were purified and hybridized onto Agilent G4851B SurePrint G3 Human Gene Expression 8*60 K v2 microarrays™, allowing full coverage of the human transcriptome (Agilent Technologies, Waldbronn, Germany). The slides were washed and scanned on an Agilent G2505C™ microarray scanner with a 3 μm resolution and data were extracted using Agilent Feature Extraction software version 11.0, as described previously by Chézeau et al. (2018) [33]. The experiments were performed according to MIAME standards [34]. The microarray data were uploaded to the NCBI Gene Expression Omnibus database [35], where they are accessible under the GEO series accession number GSE203572, [36].

### 2.7. Statistical and Functional Analyses of Microarray Data

The data were quantile-normalized using Solo software (Institut de Génétique et de Biologie Moléculaire et Cellulaire, Strasbourg) [37]. The genes displaying differential expression between control and exposed groups were identified using a method based on Fold Change Rank Ordering Statistics (FCROS; Dembele and Kastner, 2014) [38]. The genes for which the fold change (FC) for exposed vs. matched controls was at least 1.5 in either direction and with f-values ≤|0.01| were considered as significantly differentially expressed. The details and functional information on genes differentially expressed were obtained using the GeneCards™ database [39]. A functional Gene Ontology (GO) analysis of the differentially expressed genes was performed using v6.8 of the Database for Annotation, Visualization, and Integrated Discovery (DAVID) [40]. This tool was used to extract the biological meaning from a long list of differentially expressed genes and to identify biological functions possibly altered following exposure to rough Y-TZP discs. Genes that share similar functions are clustered based on biological processes, cellular components, or molecular functions. In this study, we mainly focused on GO BPs belonging to annotation clusters with an enrichment score (Z score) >1.3 [41]. We also examined significantly enriched KEGG (Kyoto Encyclopedia of Genes and Genomes) biological pathways (*p* < 0.05), a collection of manually drawn pathway maps representing knowledge on molecular interactions and reaction networks [33].

Transcript–transcript interactions were investigated using the Search Tool for the Retrieval of Interacting Genes/Proteins (STRING) [42] database of physical and functional interactions v11.5 [43], as described previously by Chézeau et al. (2019) [44]. The network nodes represent transcripts and the edges represent transcript–transcript associations. The line color of the network edges indicates the type of interaction. Thus, known interactions are represented as the light blue line (this color indicates the presence of database evidence) and purple line (experimental evidence); predicted interactions as the green line (neighborhood evidence), red line (fusion evidence), and blue line (co-occurrence evidence); and other interactions as the yellow line (text mining evidence), black line (co-expression evidence), and violet line (protein homology). The analysis section gives some brief statistics for the inferred network, such as the numbers of nodes and edges. A small PPI enrichment *p*-value indicates that the nodes are not random and that the observed number of edges is significant.

## 3. Results

### 3.1. Morphology of ZrO_2_ and Y-TZP Discs

The morphologies of rough and smooth surfaces of ZrO_2_ and Y-TZP discs without cells were assessed by SEM. Discontinuous parallel striations created by the abrasive polishing cloth with a particle size of 80 µ were observed on the rough surfaces of ZrO_2_ discs (Figure 1A,B). These reveal the brittle character of zirconia, which instead of deforming plastically, breaks cleanly under stress. The rough surfaces of Y-TZP discs also present discontinuous parallel striations, which are less pronounced (Figure 2A,B). Concerning the smooth surfaces, the ZrO_2_ discs present irregular asperities forming microcavities, contrary to Y-TZP discs, which have very few defects (Figure 1C,D, Figure 2C,D and Figure 3A). These observations suggest that the addition of Y_2_O_3_ strengthens and stabilizes the zirconia, reducing the risk of failure under mechanical stress.

### 3.2. Measurement of Metabolic Activity of Saos-2 Cells in Contact with Smooth and Rough Surfaces of ZrO_2_ and Y-TZP Discs

The mitochondrial dehydrogenase activity levels of Saos-2 cell exposed to ZrO_2_ and Y-TZP discs with mirror-polished and rough surfaces were evaluated using the WST-1 assay at 24 h, 72 h, and 96 h of exposure. Similar metabolic activity levels were observed between control and exposed cells, which increased over time. Moreover, similar metabolic activity levels were observed between ZrO_2_ and Y-TZP discs with rough surfaces on the one hand (Figure 4) and between ZrO_2_ and Y-TZP with mirror-polished surfaces on the other hand (Figure 5). Different controls were used to exclude potential interactions between the reagents, the culture medium, and the discs.

### 3.3. Morphology of Saos-2 Cells Exposed to ZrO_2_ and Y-TZP Discs

The adhesion of Saos-2 bone cells on rough and mirror-polished surfaces of ZrO_2_ and Y-TZP discs was evaluated via SEM. The cells’ orientation seemed to be slightly influenced by the surface conditions independently of the nature of the discs (ZrO_2_ or Y-TZP). Indeed, on rough surfaces, the cells seemed to be slightly more organized, according to the underlying striations (Figure 6A,B and Figure 7A,B), whereas the cells’ orientation appeared more random and anarchic on smooth surfaces (Figure 6C,D and Figure 7C,D). Most cells showed an elongated and flat morphology, while some were rounded. Moreover, important cell adhesion and spreading were noticed regardless of the type of surface or material, resulting in the formation of an important confluent cell layer. The deployment of a multitude of pseudopodia as well as a vast network of spicules and microfilaments was observed on the surfaces of the cells at high magnification, which were essential for their adhesion (Figure 6 and Figure 7B,D, ×1000).

### 3.4. Transcriptomic Analysis

The analysis of Saos-2 cells exposed to the rough surfaces of Y-TZP discs for 96 h revealed 133 differentially expressed probes on a total of 62,976, probes (full coverage of the human transcriptome); 74 probes were upregulated and 59 downregulated. This list of probes corresponds to 110 differentially expressed transcripts with an FC ≥ |1.5| and an f-value ≤ |0.01|. Among these transcripts, 25 are unknown (such as ENST, LOC, and XLOC). Thus, 85 transcripts encoding 85 known genes were identified as differentially expressed in this study. Most of these transcripts were slightly up- or downregulated. Only 12 transcripts had an FC ≥ |2|, namely *IFI27, IFI44L, MT1E, MX1, MYOD1, OAS1, OAS2, RSAD2, SOST, TRIM14,* and 2 unknown genes (*ENST00000450667* and *XLOC_008559*), which were all upregulated (Appendix A) and are mainly involved in the immune system process (Table 1). We were also interested in the 10 most downregulated transcripts in the initial list of 133 deregulated probes, namely *ATHL1, EXOC7, GZMB, H19, IGFN1, LSP1, MIR143HG, NEAT1, SAPCD1, WDR90* (Table 2). These transcripts are involved in inflammatory and defense responses (to the virus), in the metabolic process (such as proteolysis and apoptosis), and in cell motility (cytoskeleton organization and adhesion).

The second step consisted of injecting the list of 133 differentially expressed probes into the DAVID database. The analysis highlighted 11 enriched GO BPs in exposed Saos-2 cells (Z score > 1.3) (Figure 8). The most enriched GO BP areas are related to the “response to metal ions”, including 7 upregulated transcripts (*MT2A, MT1A, MT1L, MT1M, MT1X, MT1B,* and *MT1E*) that encode MT, a family of proteins acting as antioxidants that is involved in the detoxification of heavy metals, but also in bone metabolism and remodeling [45]. The other main enriched GO BP involves the cellular “immune response” that is effective against viruses and proteolysis, a process involved in the immune response and the elimination of invasive pathogens. These GO BPs include the largest number of deregulated transcripts, such as interferon-induced proteins *IFI27* and *IFI44L; MT1E, MX1,* and *MYOD1;* oligoadenylates synthetases *OAS1* and *OAS2;* and *RSAD2*, which are 8 of the most highly upregulated transcripts in our entire list (Table 1). *BST2, CD86, CXCL8 (IL-8), MX2, MT2A, IFIT1, OASL,* and *CTSO* are all upregulated. On the other hand, 3 transcripts involved in the proteolysis are downregulated: matrix metalloproteinase *MMP13, ADAMTSL5,* and *ADAMTS10*. Interestingly, *MMP13, ADAMTSL5,* and *ADAMTS10* are also involved in extracellular matrix remodeling with collagens *COL7A1, COL27A1,* and *SOST*. Surprisingly, these transcripts are all downregulated excepted *SOST. SOST* encodes sclerostin, a protein expressed by osteocytes with catabolic effects on bone, which antagonizes BMP signaling directly by inhibiting bone morphogenetic protein 7 (BMP7) secretion [46].

These 11 enriched GO BPs are related to 2 main KEGG pathways involved in mineral absorption and the immune and defense responses against viruses (Figure 8). The mineral absorption metabolic pathway is very significantly enriched, with the lowest *p*-value. Six upregulated transcripts are involved in this pathway and encode MT: *MT1A, MT1B, MT1E, MT1M, MT1X,* and *MT2A* (Table 3). The second enriched metabolic pathway correlates with the immune response, more specifically antiviral-type defense involving interferons, including “influenza A”, “measles”, and hepatitis C” KEGG pathways with 5, 4, and 4 upregulated transcripts, respectively. These transcripts include *CXCL8*, *CSNK2A1*, *IFIT1, MX1*, *OAS1, OAS2,* and *RSAD2.*

Finally, an analysis of the transcript–transcript interactions using the STRING database (v11.5) highlighted 2 main independent networks (Figure 9). The first one linked to mineral absorption is mainly composed of transcripts encoding MT. The second related to immune response contains the largest number of transcripts that encode interferon-induced proteins, the chemokine CXCL8, a major mediator of the inflammatory response, as well as other immune and defense proteins. Interestingly, 2 small networks involved in extracellular matrix remodeling were noticed: *ADAMTSL5* connected with *ADAMTS10* and *COL27A1* with *COL7A1*. These data correlate with David’s analysis. *MMP13* is connected with *CXCL8* and the immune response network on one hand, and with *SOST* on the other hand. Four other small networks were observed: *ASIC1* and *ASIC2* were downregulated, encoding acid-sensing ion channels; *ZNF692* and *CCNL2* (*Cyclin L2*) involved in the regulation of transcription by RNA polymerase II; *SPATA22* and *CTCFL* involved in the cell cycle; and *LIN7A* and *BAIAP2*. Interestingly, small PPI enrichment *p*-values were noticed, indicating that the nodes are not random and that the observed number of edges is significant.

## 4. Discussion

This study provides detailed insights concerning the biocompatibility of ZrO_2_ compared to Y-TZP discs, as well as the influence of the surface topography, through the combination of a conventional toxicological assay, morphological observations, and a transcriptomic analysis performed on Saos-2 bone cells, which are currently used in dental implant studies. The objective was to determine the best combination and parameters for optimal osseointegration with minimal risk of rejection in patients.

Currently, titanium and its alloys remain the reference materials in oral implantology. However, studies have reported the presence of titanium (Ti) particles around dental implants (released from type 4 titanium alloy) and inside epithelial cells, connective tissue, macrophages, and bone [1]. The debris released from the degradation of dental implants has cytotoxic and genotoxic potential for peri-implant tissues and can provoke an inflammatory response and osteolysis that could lead to dental peri-implantitis, the main cause of implant failure [31,47,48,49]. Concerning TiAl6V, the possibility of vanadium and aluminum release limits its use. Indeed, vanadium has a cytotoxic effect, while aluminum has significant neurotoxic effects and can induce Alzheimer’s disease, bone fragility, and local inflammation [50,51]. Hypersensitivity reactions are also reported.

In this context, the development of alloys free of toxic elements and inert in the oral environment is essential. Thus, zirconia dental implants and their alloys have emerged as an alternative to titanium. Based on its superior biomechanical properties compared to other ceramics, zirconia presents many advantages, such as its esthetic outcomes, biocompatibility, and resistance to corrosion [21]. Y-TZP is the most widely used and robust variant obtained via the addition of Y_2_O_3_ on pure zirconia for stabilization [19].

In this study, we first focused on the proliferation of Saos-2 cells exposed to rough and mirror-polished surface states of ZrO_2_ and Y-TZP discs. Our results suggest that cell proliferation is neither impacted by the nature of the material, by the surface topography, nor by the addition of Y_2_O_3_ to zirconia as a stabilizing agent. These observations correlate with the excellent biocompatibility of ZrO_2_ and its alloys often reported in the literature [8,9,10]. For instance, a large clinical study reported promising results concerning the survival rate (95.6%) and marginal bone loss of zirconia dental implants 12 months after implantation [52]. A recent study highlighted the lack of toxicity of a large volume of Y_2_O_3_ (more than 8 mol%) in L929 fibroblast-like cells and mouse bone marrow-derived mesenchymal stem cells [53].

On the other hand, the addition of Y_2_O_3_ to zirconia does not seem to have any impact on the cell morphology compared to ZrO_2_, as shown by the SEM observations. Indeed, similar cell spread and adhesion were noticed on the surfaces of ZrO_2_ and Y-TZP discs with the presence of an important confluent cell layer. In contrast, the surface topography seems to impact the cell morphology. A slight cellular alignment along the underlying striations was observed on the rough surfaces, contrary to a more random and anarchic organization on the smooth surfaces. According to the literature, rough-surfaced implants promote osseointegration, favoring both bone anchoring and biomechanical stability [28,29,54]. The precise role of the surface chemistry and topography in the early events of dental implant osseointegration remains poorly understood [28,32]. Therefore, the challenge is to find an ideal surface roughness that combines optimal bone fixation with minimal bacterial retention. In this context, the advantage of Y-TZP is its excellent biocompatibility and its ability to limit bacterial proliferation [20,21].

These different observations in terms of the mechanical properties, proliferation, and cell morphology, as well as the excellent biocompatibility and capacity to limit bacterial proliferation, led us to focus more precisely on the rough surfaces of Y-TZP discs at the molecular level. Thus, a transcriptomic analysis was performed on Saos-2 cells exposed to rough Y-TZP for 96 h to investigate the potential modification of gene expression and biological processes compared to control cells. Interestingly, only 110 transcripts were differentially expressed out of a total of 62,976 probes analyzed, which correlates with the excellent biocompatibility of Y-TZP commonly reported in the literature.

The main enriched (Z-Score > 1.3) biological processes (GO BPs) modified in Saos-2 cells exposed to rough Y-TZP discs are involved in 2 metabolic pathways, which are also enriched (*p* < 0.05) in relation to mineral absorption and the immune response, more precisely the interferon antiviral response. The metabolic pathway concerning mineral absorption includes 6 transcripts, *MT1A, MT1B, MT1E, MT1M, MT1X,* and *MT2A,* which are all upregulated and encode MT, a family of proteins involved in heavy-metal sequestration and detoxification, acting as antioxidants [55]. Interestingly, these proteins are also involved in bone metabolism, remodeling, and development, and the progression of dental caries [45,56]. A recent study highlighted the low expression levels of various *MT* transcripts, including our 6 upregulated transcripts, in Down’s syndrome patients with periodontitis and implant rejection [45]. In this study, all metallothionein transcripts present in human cells (on a total of 62,976 probes) were upregulated (*MT1L, MT1X, MT1B, MT1M, MT1A, MT1HL1, MT1E, MT2A*). Another study mentioned the ability of MT to protect mouse bone marrow stromal cells against the oxidative-stress-induced inhibition of osteoblastic differentiation [57]. Thus, we can reach an important hypothesis about the involvement of MT in the early stage of implant osseointegration, namely that it promotes osteoblasts [45,58]. This hypothesis needs to be confirmed by further investigations. Moreover, it could be interesting to determine if the expression of *MT* transcripts is influenced by the nature and surface topography of the implants, as well as to focus on the expression of MT at the protein level by performing Western blotting. Indeed, only a few recent studies have focused on the role of MT in dentistry and dental implants concerning bone remodeling.

On the other hand, the second enriched metabolic pathway is correlated with the immune response, more specifically the antiviral defense involving interferons. This pathway contains the largest number of deregulated transcripts (18), mainly encoding interferon-induced proteins, but also central mediators of inflammation such as the chemokine CXCL8. Therefore, the cells seem to consider rough Y-TZP as a foreign body while tolerating it, as observed previously, with efficient cell proliferation and adhesion onto the surfaces. Similarly, a previous DNA microarray analysis performed on MG-63 osteoblasts-like cells cultured on ZrO_2_ discs revealed differentially expressed genes involved in immunity, vesicular transport, and cell cycle. Thus, ZrO_2_ and Y-TZP would be able to modulate immunity, to allow them to be recognized as “self” by cells [59]. However, it is important to note that the rough Y-TZP discs upregulated a limited number of genes involved in immunity (18), the expression of which should probably normalize in the long run. In conclusion, rough Y-TZP is very well tolerated by Saos-2 cells.

Then, 6 restricted transcript–transcript interaction networks were identified using the STRING database. The 2 most interesting are related to extracellular matrix remodeling, with a slight downregulation of *COL7A1, COL27A1, ADAMTSL5,* and *ADAMTL10*. *MMP13*, which encodes a metalloproteinase and proteins involved in bone remodeling, is also downregulated. The extracellular matrix proteins are involved in osseointegration and bone resorption, remodeling, and repair, along with the regulation of osteoblasts and osteoclasts [60,61]. Thus, the downregulation of only 5 transcripts could be related to a very slight disturbance in bone remodeling. In this context, we wanted to deepen our analysis by focusing on the genes involved in bone remodeling, ossification, osteogenic differentiation, and osseointegration. The dentistry literature is well documented and the main specific markers frequently reported are: *RUNX family transcription factor 2 (RUNX2*), *alkaline phosphatase (ALP)*, *osteopontin (OPN)* or *SPP1*, *osteocalcin (OCN*) or *BGLAP, osteonectin (ON*) or *SPARC*, *bone morphogenetic protein-2 (BMP-2)*, *type 1 collagen (COL1), transforming growth factor-b (TGF-b),* the early osteogenic marker *Sp7 transcription factor (Sp7)* the late osteogenic marker *bone sialo protein* (*BSP* or *IBSP*), and the osteocyte differentiation marker *sclerostin (SOST)* [62,63,64,65]. Surprisingly, in this study none of these markers were deregulated except *SOST*. One hypothesis is that the deregulation of transcripts involved in bone remodeling occurs later than 96 h. Thus, it would be interesting to analyze these transcripts after a long time of exposure and to analyze the protein expression levels using a Western blot or ELISA assay (OPN, APL). Interestingly, *SOST,* which was upregulated in this study, encodes sclerostin, an osteocyte differentiation marker. This protein is expressed by osteocytes and is a negative regulator of bone formation, which antagonizes BMP signaling directly by inhibiting BMP7 secretion, and also stimulates bone resorption. Although *BMP7* expression was not modified in this study, the SOST protein could inhibit BMP7 expression at the protein level. In periodontitis, the synthesis of sclerostin increases related to the activation of the inflammatory cascade [46,55,66]. Thus, it could be interesting to determine whether the overexpression of *SOST* persists for a longer time and correlates with the overexpression of SOST protein, or if the regulatory mechanisms lead to the normalization of protein expression (some miRNA were downregulated in our list).

Taken together, the transcriptomic analysis confirmed the excellent biocompatibility of rough Y-TZP, with only 110 transcripts being deregulated, which are mainly involved in mineral absorption and the immune response. These results correlate with the WST1 assay and morphological observations. Interestingly, few of these deregulated transcripts are involved in cell motility, anchoring, or proliferation. *CORO6* and *mir143* were downregulated, encoding proteins and microRNAs involved in actin filament organization and cell migration. *BAIAP2* was downregulated and encodes a protein involved in lamellipodia and filopodia formation. *WDR90* was also downregulated and encodes a protein involved in cilium assembly. *ITGB2* was upregulated and encodes an integrin, which belongs to a family of proteins that participate in cell adhesion. *SAMD9* was slightly upregulated and encodes a protein that may play a role in regulating cell proliferation and apoptosis. *ANXA10* was upregulated and is involved in the regulation of cellular growth (GeneCards). Finally, *ALDH1A3* was downregulated and encodes a protein that seems to participate in extracellular matrix organization and cell adhesion [67].

The strength of this study is that very few transcriptomic analyses have been performed on dental implants; one previous study focused on ZrO_2_ discs but none have focused on Y-TZP to our knowledge [45,59,68,69]. In this context, it could be interesting to analyze the gene expression at 24 h to observe early deregulations and their possible persistence or normalization at 96 h and over longer periods (7 and 21 days) of exposure to rough Y-TZP discs. Indeed, wound healing, specifically around a dental implant, is based on 4 coordinated and sequential phases of reparation: hemostasis (minutes to hours), inflammation (hours today), proliferation (days to weeks), and remodeling (up to 3 weeks or last for days) [70]. Thus, the deregulation of specific transcripts corresponding to these 4 steps could be observed at different times. Moreover, it could be interesting to perform a comparative transcriptomic study between rough and mirror-polished Y-TZP samples to observe a potential influence of the surface topography on the gene expression and to better understand the underlying mechanisms.

## 5. Conclusions

This study aimed to assess the influence of the nature of the materials and the surface topography on bone cells, which influence osseointegration. The morphological observations highlighted an important adhesion of cells onto ZrO_2_ and Y-TZP surfaces at 96 h. The deployment of cellular pseudopods and the formation of an important confluent layer are encouraging for the osseointegration process. Moreover, the cell proliferation rates are similar between control cells, ZrO_2_, and Y-TZP, regardless of the surface topography. Finally, the transcriptomic analysis performed on cells exposed to the rough surface of Y-TZP for 96 h confirmed the excellent biocompatibility of rough Y-TZP discs, with only 110 deregulated transcripts in the entire human transcriptome. Therefore, Y-TZP is a serious candidate for implantology in general. However, further biocompatibility and biomechanical studies are needed to position Y-TZP as a reference material in oral implantology and to find the best roughness level for this material.

## Figures and Tables

**Figure 1 materials-15-04655-f001:**
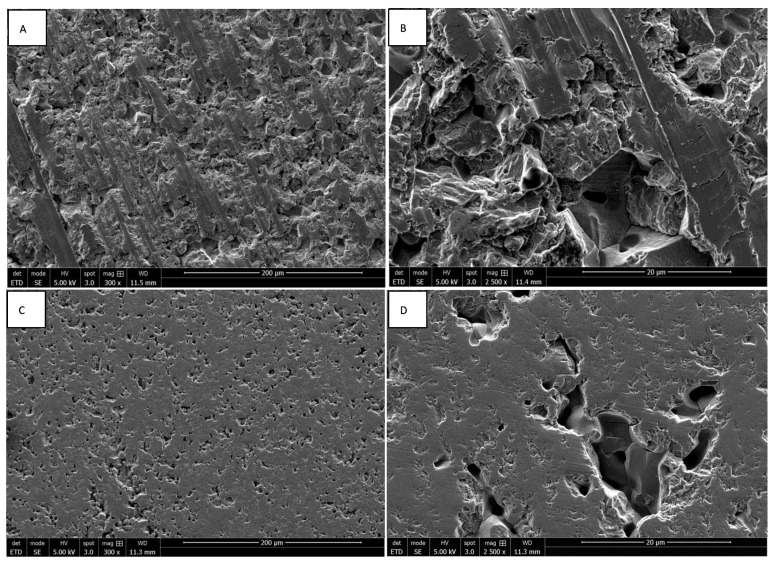
SEM images of rough ((**A**) magnification 300×; (**B**) magnification 2500×) and smooth surfaces ((**C**) magnification 300×; (**D**) magnification 2500×) of ZrO_2_ discs.

**Figure 2 materials-15-04655-f002:**
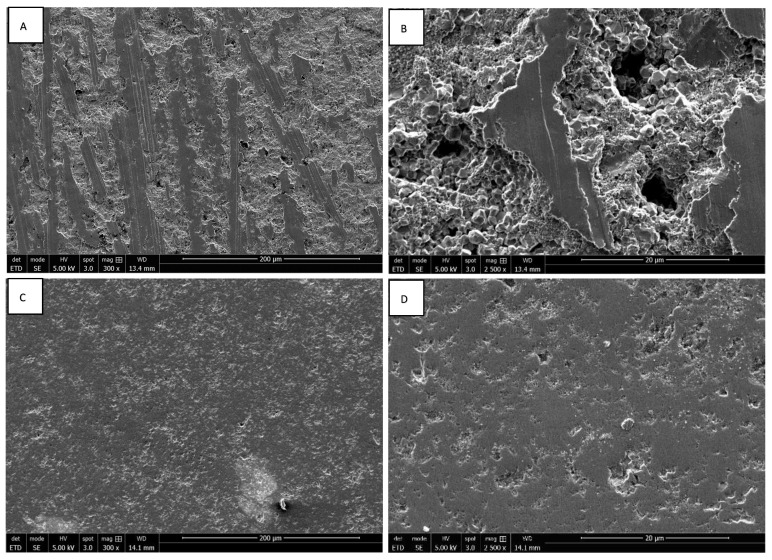
SEM images of rough ((**A**) magnification 300×; (**B**) magnification 2500×) and smooth surfaces ((**C**) magnification 300×; (**D**) magnification 2500×) of Y-TZP discs.

**Figure 3 materials-15-04655-f003:**
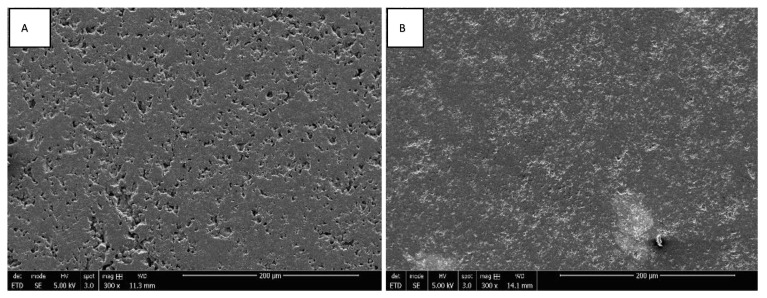
SEM images of smooth surfaces of ZrO_2_ ((**A**) magnification 300×) and Y-TZP ((**B**) magnification 300×) discs.

**Figure 4 materials-15-04655-f004:**
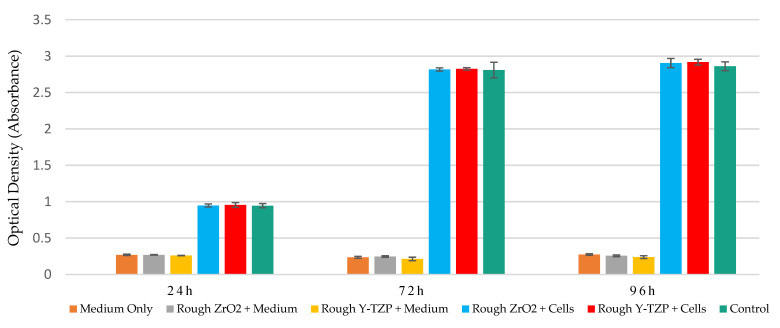
Metabolic activity monitoring (WST1 assay) of Saos-2 cells exposed to rough surfaces (Ra = 1 µm) of ZrO_2_ and Y-TZP for 24 h, 72 h, and 96 h (N = 4). ANOVA: Analysis of one-factor variances (ZrO_2_ vs. Y-TZP vs. controls). Not Significant at 24 h, 48 h, and 96 h.

**Figure 5 materials-15-04655-f005:**
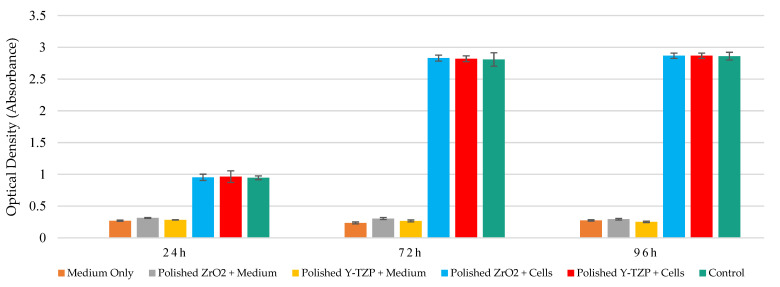
Metabolic activity monitoring (WST1 assay) of Saos-2 cells exposed to mirror-polished surfaces (Ra = 0.01 µm) of ZrO_2_ and Y-TZP for 24 h, 72 h, and 96 h (N = 4). ANOVA: Analysis of one-factor variances (ZrO_2_ vs. Y-TZP vs. controls). Not Significant at 24 h, 72 h, or 96 h.

**Figure 6 materials-15-04655-f006:**
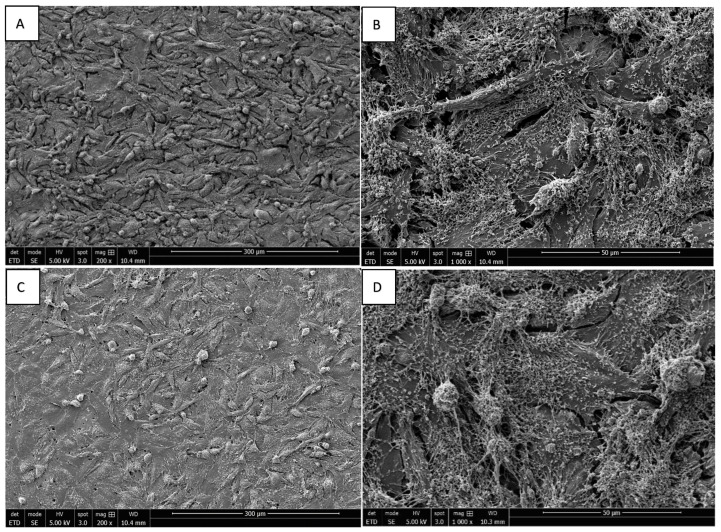
SEM image of Saos-2 cells on rough ((**A**) magnification 200×; (**B**) magnification 1000×) and mirror-polished ((**C**) magnification 200×; (**D**) magnification 1000×) surfaces of ZrO_2_ discs at 96 h of incubation.

**Figure 7 materials-15-04655-f007:**
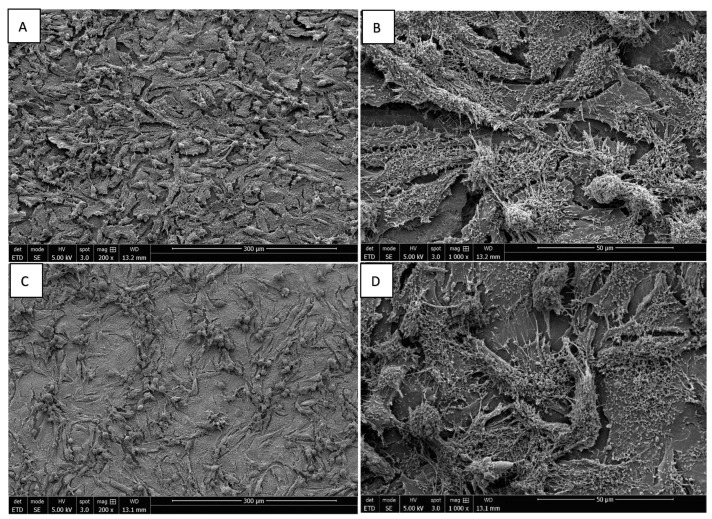
SEM image of Saos-2 cells on rough ((**A**) magnification 200×; (**B**) magnification 1000×) and mirror-polished ((**C**) magnification 200×; (**D**) magnification 1000×) surfaces of Y-TZP discs at 96 h of incubation.

**Figure 8 materials-15-04655-f008:**
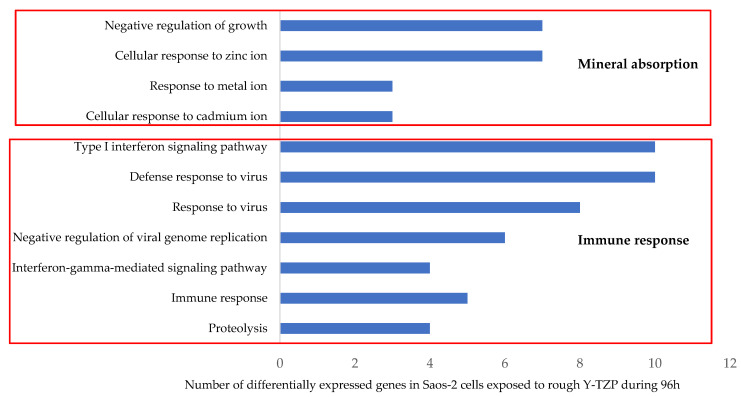
Enriched Gene Ontology Biological Processes (GO BPs) modified in Saos-2 cells exposed to rough surfaces (Ra = 1 µm) of Y-TZP discs for 96 h (determined through DAVID Bioinformatics Resources v6.8). (Z score > 1.3).

**Figure 9 materials-15-04655-f009:**
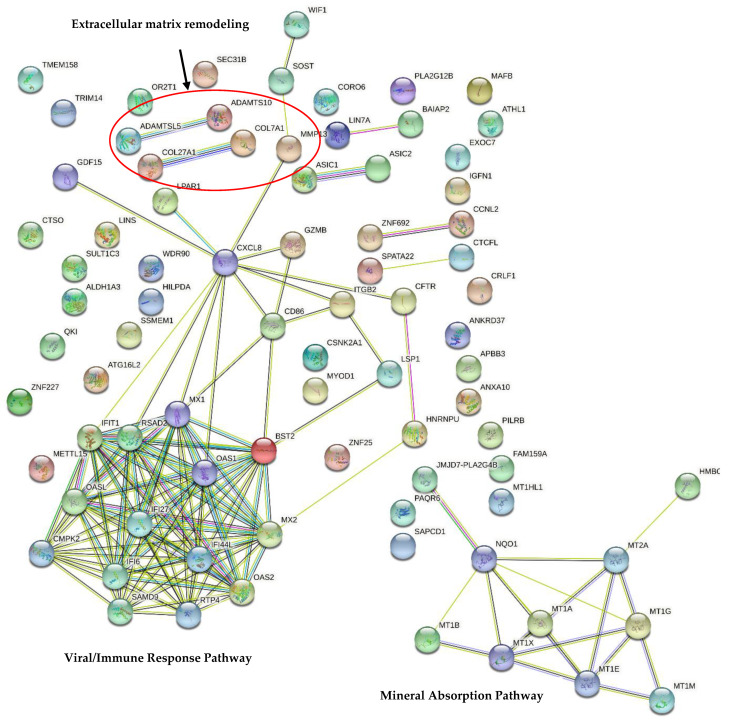
The STRING network of gene–gene interactions in Saos-2 cells exposed to the rough surfaces (Ra = 1 µm) of Y-TZP discs for 96 h (STRING database version 11.5). A list of all differentially expressed genes in Saos-2 cells exposed to the rough surfaces of Y-TZP discs for 96 h (FC ≥ |1.5, |f-value ≤ |0.01) (Appendix A) was injected into STRING database v11.5. The 2 significantly enriched KEGG pathways (determined using the DAVID database) are represented in bold. Legend: Network nodes represent proteins, edges represent protein–protein associations (red line: the presence of fusion evidence; green line: neighborhood evidence; blue line: co-occurrence evidence; purple line: experimental evidence; yellow line: text mining evidence; light blue line: database evidence; black line: co-expression evidence).

**Table 1 materials-15-04655-t001:** Saos-2 cells exposed on the rough surfaces of Y-TZP discs for 96 h (f-value ≤ |0.01).

Gene Name	FC2	f-Value	Gene Ontology Biological Process (GO BP)
ENST00000450667	2.04	0.997	/
*IFI27*	2.47	0.997	Immune system process (cellular antiviral response)
*IFI44L*	2.36	0.997	Involved in immune response (defense response to the virus)
*MT1E*	2.24	0.997	Cellular response to metal ion
*MX1*	2.74	0.998	Immune system process (cellular antiviral response)
*MYOD1*	2.36	0.998	Involved in muscle organ development
*OAS1*	2.45	0.997	Immune system process (cellular antiviral response)
*OAS2*	2.01	0.997	Immune system process (cellular antiviral response)
*RSAD2*	2.18	0.997	Immune system process (cellular antiviral response)
*SOST*	2.33	0.997	Involved in ossification
*TRIM14*	2.03	0.996	Immune system process (protein ubiquitination)
*XLOC_008559*	2.11	0.997	/

**Table 2 materials-15-04655-t002:** The most downregulated transcripts in Saos-2 cells exposed on the rough surfaces of Y-TZP discs for 96 h (f-value ≤ |0.01).

Gene Name	FC2	f-Value	Gene Ontology Biological Process (GO BP)
*ATHL1 (PGGHG)*	0.57	0.003	Metabolic process
*EXOC7*	0.59	0.004	Involved in regulation of entry of bacterium into the host cell (exocytosis)
*GZMB*	0.51	0.003	Proteolysis (apoptotic process)
*H19*	0.58	0.003	Involved in the cellular response to virus
*IGFN1*	0.59	0.003	Involved in homophilic cell adhesion via plasma membrane adhesion molecules
*LSP1*	0.57	0.003	Defense response
*MIR143HG*	0.59	0.003	Involved in actin cytoskeleton organization
*NEAT1*	0.57	0.003	Involved in positive regulation of inflammatory response
*SAPCD1*	0.6	0.004	/
*WDR90*	0.59	0.004	Cell projection organization (cilium assembly)

**Table 3 materials-15-04655-t003:** Cells exposed to the rough surfaces (Ra = 1 µm) of Y-TZP discs for 96 h (*p*-value < 0.05).

KEGG	Number of Genes	Genes	*p*-Value
Mineral absorption	6	*MT2A, MT1A, MT1M, MT1X, MT1B, MT1E*	2.386 × 10^−6^
Influenza A	5	*CXCL8, RSAD2, OAS1, OAS2, MX1*	0.011
Measles	4	*CSNK2A1, OAS1, OAS2, MX1*	0.029
Hepatitis C	4	*CXCL8, OAS1, OAS2, IFIT1*	0.029

## Data Availability

Data sharing is not applicable.

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
