# Peer review of "Biocompatibility of ZrO2 vs. Y-TZP Alloys: Influence of Their Composition and Surface Topography"

_materials, 2022, doi:10.3390/ma15134655_

Round 1

Reviewer 1 Report

1) Show some important numerical values of the results in the abstract and conclusions.

2) Write the novelty of the work in 5 nice sentences.

3) Give examples for each of these properties and write each of its advantages: exceptional mechanical properties, esthetic outcomes, biocompatibility, and resistance to corrosion.

4) If only a few investigations have been carried out on the influence of 93 ZrO2 on gene expression profile and none on Y-TZP, please prepare a table and summarize the key conditions and results obtained by previous researchers.

5) Compare the Mitochondrial dehydrogenase activity results with other works (at least 5-6 references and the conditions used).

6) Please make sure that the graphs are prepared as follows (general guideline):

- X and Y axis lines are black in colour and also the numbers and text in the axis are in black colour. At present, they are in GREY.

- Error bars are provided

- Units are written in a consistent format

- Spacing errors are avoided 

- Delete the outer border in the figure (box type external border)

- Use Arial font in the x and y axis

- In the legends, there should not be any formatting mistakes or spacing mistakes

- Please make sure that there are no grid lines

- Major tick marks should be outside (check your excel file settings and formatting)

7) The role of spicules and microfilaments should be discussed in DETAIL.

8) Divide the discussion into several sub-sections depending on the specific objectives of this work.

9) Write the practical applications and future research prospects of this work before the conclusions section, as a separate section (max 200 words).

10) Write the specific objectives of this work clearly in the last paragraph of the introduction (at least 5-6 sentences).

11) Check REF formatting manually and avoid formatting errors.

Author Response

Reply to reviewer 1

  • Show some important numerical values of the results in the abstract and conclusions.

Answer 1 : The most important results are not numerical values but have been recorded in the summary and conclusion

  • Write the novelty of the work in 5 nice sentences.

Answer 2 :There are few studies on the genetic influence of the components of various materials used in implantology. The novelty of this work is the provision of a body of information on the genetic determinism of yttriated zirconia on human cells in order to gain insight into the genomic/metabolic impact of the addition of yttrium oxide as a zirconia stabiliser.

  • Give examples for each of these properties and write each of its advantages: exceptional mechanical properties, esthetic outcomes, biocompatibility, and resistance to corrosion.

Answer 3 : the properties and applications of these materials are widely known to the general public and the classic bibliography contains a large amount of information on the advantages and disadvantages of these materials, which was not the objective of this work

  • If only a few investigations have been carried out on the influence of 93 ZrO2 on gene expression profile and none on Y-TZP, please prepare a table and summarize the key conditions and results obtained by previous researchers.

 Answer 4 : The aim of this work was simply to compare two materials and not a meta-analysis of data that had already been done on the subject.

  • Compare the Mitochondrial dehydrogenase activity results with other works (at least 5-6 references and the conditions used).

Answer 5 : This comparison has already been made in the discussion section and even though the cell profiles used are varied in nature, the results are more or less the same in cases of similar operation.

6) Please make sure that the graphs are prepared as follows (general guideline):

- X and Y axis lines are black in colour and also the numbers and text in the axis are in black colour. At present, they are in GREY.

  Answer  : corrected

- Error bars are provided

Answer  : corrected

- Units are written in a consistent format

Answer  : corrected

- Spacing errors are avoided

Answer  : corrected

- Delete the outer border in the figure (box type external border)

Answer  : corrected

- Use Arial font in the x and y axis

Answer  : we leave it to the magazine to put the format of their choice

- In the legends, there should not be any formatting mistakes or spacing mistakes

Answer  : corrected

- Please make sure that there are no grid lines

Answer : there is none

- Major tick marks should be outside (check your excel file settings and formatting)

Answer : please specify which file it is

  • The role of spicules and microfilaments should be discussed in DETAIL.

Answer 7 : please specify the relevant section

  • Divide the discussion into several sub-sections depending on the specific objectives of this work.

 Answer 8 : It was planned to separate the discussion into several sections, but as the results are linked, it was realised that the presence of several sections made the ideas less coherent.

  • Write the practical applications and future research prospects of this work before the conclusions section, as a separate section (max 200 words).

   Answer 9 :The future research perspectives of this work have already been mentioned in the last discussion section

  • Write the specific objectives of this work clearly in the last paragraph of the introduction (at least 5-6 sentences).

Answer 10 :7 lines of text are devoted to the objectives of this work which were clearly explained and detailed in the last section of the introduction 

10) Check REF formatting manually and avoid formatting errors.

Answer 11 : References vitrified, but please specify the wrong reference numbers

Reviewer 2 Report

The goal of this study is to compare the biocompatibility of yttria zirconia (Y-TZP) discs with rough (Ra= 1 m) and smooth (Ra= 0 m) surface conditions to pure zirconia (ZrO2) discs using a combination of traditional toxicological assays, morphological observations, and transcriptomic analysis on an in vitro model of human Saos-2 bone cells. Through the review, I think this is a very nicely written paper, and the author analyzed and discussed the results in detail and correctly. The analysis developed in this paper is correct and the obtained results are interesting. The paper has sufficient novelty which covers the scope of the Materials journal. Therefore, the manuscript may consider for publication in the Materials journal after responding to the following comments and major revising the manuscript properly.

1. The Abstract is too long to follow. The Abstract mainly contains an enumeration of methods, but there is no general information on the results achieved. The abstract should summarize the findings of the work.

2. Various materials, chemicals, and characterization devices are used in this experiment. Please write all of these materials and devices model, and origin/country clearly.

3. The quality of all Figure 9 is too low, please increase the quality. If necessary, redraw.

4. In all SEM images, the length bar is not clear (like 200 μm white bar in figure 1a). Please add an extra clear horizontal bar in all SEM images of Figures 1, 2, 6, and 7.

5. Authors are suggested to use more references from the recent past and recommend to cite the following references in the introduction section. DOI: 10.1016/j.apmt.2021.101104; 10.1039/D2RA00006G; 10.1021/acsaelm.1c00703.

6. Also, check the typos throughout the manuscript during revision submission.

Author Response

Reply to reviewer 2

The goal of this study is to compare the biocompatibility of yttria zirconia (Y-TZP) discs with rough (Ra= 1 m) and smooth (Ra= 0 m) surface conditions to pure zirconia (ZrO2) discs using a combination of traditional toxicological assays, morphological observations, and transcriptomic analysis on an in vitro model of human Saos-2 bone cells. Through the review, I think this is a very nicely written paper, and the author analyzed and discussed the results in detail and correctly. The analysis developed in this paper is correct and the obtained results are interesting. The paper has sufficient novelty which covers the scope of the Materials journal. Therefore, the manuscript may consider for publication in the Materials journal after responding to the following comments and major revising the manuscript properly.

Thank you for your keen interest in our work

  1. The Abstract is too long to follow. The Abstract mainly contains an enumeration of methods, but there is no general information on the results achieved. The abstract should summarize the findings of the work.

Answer 1 : The summary has been modified and reduced

  1. Various materials, chemicals, and characterization devices are used in this experiment. Please write all of these materials and devices model, and origin/country clearly.

Answer 2 : Missing references have been added

  1. The quality of all Figure 9 is too low, please increase the quality. If necessary, redraw.

Answer 3 : The journal requires us to compress the handwritten data when submitting, which is why some of the images are of poor quality. i have changed all the images and they are of better quality now

  1. In all SEM images, the length bar is not clear (like 200 μm white bar in figure 1a). Please add an extra clear horizontal bar in all SEM images of Figures 1, 2, 6, and 7.

Answer 4 : The length bar is not clear due to poor image quality as explained above. the images have been changed, the length bar is now clearly visible

  1. Authors are suggested to use more references from the recent past and recommend to cite the following references in the introduction section. DOI: 10.1016/j.apmt.2021.101104; 10.1039/D2RA00006G; 10.1021/acsaelm.1c00703.

Answer 5 : Recent references have been used, but we do not understand the relevance of a rare earth study to our study.

  1. Also, check the typos throughout the manuscript during revision submission.

Answer 6 : Faults have been checked
